# Intra-abdominal hypertension and abdominal compartment syndrome in the critically ill liver cirrhotic patient–prevalence and clinical outcomes. A multicentric retrospective cohort study in intensive care

**Rui Pereira[1]\*, Maria Buglevski[2], Rui Perdigoto[1], Paulo Marcelino[3], Faouzi Saliba[4], Stijn Blot[5], Joel Starkopf[2]**

**1** Hospital de Curry Cabral, Centro Hospitalar Universitário Lisboa Central, Lisbon, Portugal, **2** University of Tartu, Tartu University Hospital, Tartu, Estonia, **3** Hospital de Santa Marta, Centro Hospitalar Universitário Lisboa Central, Lisbon, Portugal, **4** AP-HP Hôpital Paul Brousse, Université Paris Saclay, Villejuif, France, **5** Ghent University, Ghent, Belgium

\* rui.pereira@mail.com

## Abstract

### Background

Liver cirrhosis and ascites are risk factors for intra-abdominal hypertension (IAH) and abdominal compartment syndrome (ACS); however, data is scarce. We aimed to determine the prevalence of IAH/ACS in a population of critically ill cirrhotic patients with acute medical illness in intensive care and to assess for risk factors and clinical outcomes.

### Methods

This was a multicentric retrospective cohort study including two general ICUs and pooled data from a multicentric study between January 2009 and October 2019.

### Results

A total of 9,345 patients were screened, and 95 were included in the analysis. Mean age was 56.7±1.3 years, and 79% were male. Liver cirrhosis etiology included alcohol in 45.3% and alcohol plus hepatitis C virus in 9.5%. Precipitating events included infection in 26% and bleeding in 21% of cases. Mean severity score MELD and SAPS II were 26.2±9.9 and 48.5±15.3, respectively, at ICU admission. The prevalence of IAH and ACS was respectively 82.1% and 23.2% with a mean value of maximum IAP of 16.0±5.7 mmHg and IAH grades: absent 17.9%, I 26.3%, II 33.7%, III 17.9%, and IV 4.2%. Independent risk factors for IAH were alcoholic cirrhosis (p = 0.01), West-Haven score (p = 0.01), and $PaO_2/FiO_2$ ratio (p = 0.02); as well as infection (p = 0.048) for ACS. Overall, 28-day mortality was 52.6% associated with higher IAP and ACS, and independent risk factors were MELD (p = 0.001), white blood cell count (p = 0.03), $PaO_2/FiO_2$ ratio (p = 0.03), and lactate concentration (p = 0.04) at ICU admission.

**Data Availability Statement:** All relevant data are within the manuscript and its Supporting Information files.

**Funding:** The authors received no specific funding for this work.

**Competing interests:** The authors have declared that no competing interests exist.

## Conclusions

This study demonstrates a very high prevalence of IAH/ACS in the critically ill cirrhotic patient in intensive care. Increased IAP and ACS were associated with severity of disease and adverse outcomes and independent risk factors for IAH were alcoholic cirrhosis, hepatic encephalopathy and PO2/FiO2 ratio, as well as infection for ACS. Early diagnosis, prevention, and treatment of IAH/ACS might improve outcome in critically ill cirrhotic patients.

## Introduction

Pathologic increases of intra-abdominal pressure (IAP) are associated with increased morbidity and mortality in the critically ill patient [1, 2]. Intra-abdominal hypertension (IAH) and abdominal compartment syndrome (ACS), with acute organ failure, are recognized as independent mortality risk factors in several clinical settings, including major abdominal surgery, ruptured abdominal aortic aneurism, major burns, major trauma, acute pancreatitis, and mechanical ventilation [1, 2]. In mixed populations of critically ill patients in Intensive Care the prevalence of IAH ranged between 31%-51% with mortality rates between 25–59% [3–5]. Furthermore, ACS occurred in 4%-8% of these patients. Acute organ failure in this setting can caused by several mechanisms including reduced abdominal perfusion pressure with impaired organ perfusion and reduced venous return resulting in ischemic acute kidney injury, liver and gut dysfunction; cardiogenic shock due to reduced venous return; direct transdiaphragmatic pulmonary compression with elevated intrathoracic pressures, impaired respiratory volumes and elevated upper vena cava venous pressure, possibly leading to elevated intracranial pressure [6].

Patients with advanced liver cirrhosis are at high risk of developing IAH/ACS [1]. Increased portal vein pressure and subsequent intra-abdominal ascites stipulate the need for symptomatic treatment [7]. Ultimately, refractory ascites leads to chronically increased IAP and the need for repeated percutaneous drainage through paracentesis [8]. There is, however, paucity of data regarding the incidence of IAH and ACS in critical cirrhotic patients with acute medical illness.

The aim of this study was to assess the prevalence of IAH/ACS, risk factors and impact on clinical outcomes in the liver cirrhosis patient admitted into the intensive care unit (ICU) with acute medical (non-surgical) illness.

## Methods and materials

The study has been approved by the Ethics Committee of both Centro Hospital Universitário Lisboa Central, E.P.E. and Tartu University Hospital. All study procedures followed the principles of the Declaration of Helsinki. The reporting of this study followed Strengthening the Reporting of Observational Studies in Epidemiology (STROBE) guidelines.

### Study design, settings and population

This was a multicentric retrospective cohort study of patients admitted into the ICU with liver cirrhosis between 2009 and 2019.

Participating centers included two ICUs and pooled data from a multicentric study on intra-abdominal infection/sepsis [9]. Center one was a general ICU, with 21 beds, specializing in liver disease at Hospital de Curry Cabral, Centro Hospitalar Universitário Lisboa Central, in

Portugal. Standard of care included IAP measurement and patients were retrospectively screened for eligibility between October 2016 and October 2019. Center two was a general ICU, with 10 beds, mostly treating emergency patients at Tartu University Hospital at Estonia. Standard of care included IAP measurement and patients were retrospectively screened for eligibility between January 2009 and December 2018. Supplemental patient enrollment was pooled from the "Abdominal Sepsis Study: Epidemiology of Etiology and Outcome" (AbSeS), a multicenter, prospective, observational, epidemiological study, from the ESICM Trials Group Project, on adult ICU patients diagnosed with intra-abdominal infection [9]. The study period length was determined by data availability while clinical guidelines remained largely unchanged regarding the management of IAH/ACS.

Patients admitted in the ICU were screened for the following inclusion criteria: age ≥18 years, first ICU admission during the index hospital stay and a diagnosis of liver cirrhosis. The definition of liver cirrhosis was determined by the presence of bridging fibrosis on previous liver biopsy or a composite of clinical signs and laboratory tests, endoscopy, and radiologic imaging [10]. The exclusion criteria were absence of IAP recordings and/or any type of surgical patient, since most surgical ICU patients may present anatomical changes to the chest and/or abdominal wall, interfering with the normal reference range for IAP. Therefore, this cohort was restricted to adult liver cirrhotic patients with acute medical (non-surgical) conditions.

### Baseline variables

Baseline variables were recorded on ICU admission day and included demographics, liver cirrhosis etiology, precipitating event of acute illness, arterial blood lactate concentration, vital organ support with vasopressors and mechanical ventilation, and the following clinical severity scores according to the original formulas: Acute Physiology and Chronic Health Evaluation II (APACHE II), Simplified Acute Physiology Score II (SAPS II), Model for End Stage Liver Disease (MELD), Model for End Stage Liver Disease Sodium (MELD-Na), Sequential Organ Failure Assessment (SOFA), and Chronic Liver Failure SOFA (CLIF-SOFA) [11–16]. Renal replacement therapy (RRT) during the ICU stay was also recorded. Acute-on-chronic liver failure (ACLF) syndrome and organ failure (OF) were defined as per the European Foundation for the Study of Chronic Liver Failure Consortium (CLIF-C) [17]. All data on patient characteristics were retrieved on site from medical records and collected in an anonymous and protected database.

### Intra-abdominal pressure assessment and clinical management

Intra-abdominal pressure assessment, definitions, and clinical management of IAH and ACS in these patients followed the updated guidelines published by the World Society of Abdominal Compartment Syndrome (WSACS) [1, 18, 19]. Accordingly, IAP monitoring was performed via trans-bladder measurement technique with zero-pressure reference point in mmHg set at the phlebostatic axis in the midaxillary line. In critically ill adult patients IAP is approximately 5–7 mmHg and IAH is classified into Grade I, IAP 12–15 mmHg; Grade II, IAP 16–20 mmHg; Grade III, IAP 21–25 mmHg; and Grade IV, IAP >25 mmHg. Abdominal compartment syndrome (ACS) was defined as a sustained IAP >20 mmHg associated with new organ dysfunction/failure. The patient's maximum IAP was determined using the single highest IAP value of all known measurements, and mean IAP was a simple mean calculation using all recorded measurements during the entire ICU stay.

### Clinical management

Clinical management of IAH/ACS in the critically ill liver cirrhosis patients included evacuating intraluminal contents and intra-abdominal space-occupying lesions, such as ascites

through paracentesis, improving abdominal wall compliance, optimizing fluid administration, and optimizing systemic and regional perfusion. This clinical management was performed by intensivists and complied with the guidelines for the treatment of precipitating events of clinical decompensation in cirrhosis and multiorgan system failure support [8, 20, 21].

## Statistical analysis

Statistical analysis described discrete variables through count and percentage and continuous variables using mean (SD) or median (Q1, Q3), where appropriate. The Pearson coefficient was used to test correlations between continuous variables. Univariate analysis of continuous variables was performed using the t-test or the Mann-Whitney U test as a non-parametric alternative where appropriate. Chi-square and Fisher's exact tests were used for categorical variables. Multivariable analysis in backward stepwise logistic regression was performed using variables with p value ≤0.10 in univariable analysis. For multivariate statistical analysis the etiology of liver cirrhosis was dichotomized to compare between "alcohol alone or combined" versus "all other causes" and, similarly, precipitant events of clinical decompensation were categorized into "infection", "bleeding" and "other". Statistical significance was defined as p-value ≤0.05. Statistical analysis was performed using IBM Statistical Package for Social Science (SPSS), version 23 (IBM Corp, North Castle, NY, US).

## Results

A total of 9,345 ICU admissions were screened, of which 554 were adult liver cirrhosis patients in their index ICU admission; and 459 were excluded either due to surgical patient type and/or absence of IAP measure. Ninety-five patients were included in this analysis. A patient flow-chart is described in Fig 1.

Baseline characteristics of patients at ICU admission and comparison between survivors and non-survivors at 28 days are shown in Table 1. Liver cirrhosis etiology included alcohol in 43 (45.3%) cases, alcohol plus Hepatitis C (HCV) in 6 (9.5%), HCV alone in 6 (6.3%), and other causes such as NASH, alfa-1-antitripsin deficiency, and haemochromatosis in 20 (21.0%) cases. In 17 patients (17.9%), the cause remained unspecified. Comorbidities were reported in 66 patients with a mean Charlson Comorbidity Index (CCI) of 5.4±2.1. The comorbidities included cirrhosis with clinical signs of portal hypertension (86.4%), diabetes mellitus with and without end-organ damage (9.1% and 22.7%, respectively), solid cancer with metastasis (4.5%) and without metastasis (13.6%), hematologic cancer (1.5%) and chronic kidney disease (15.2%). Precipitating events of decompensated cirrhosis leading to ICU admission included infection (26.2%), bleeding (21.2%), alcohol intoxication (4.2%), acute kidney injury, hepatic encephalopathy and refractory ascites (each 3.2%). Additionally, at ICU admission, organ failures included cardiovascular (72,7%), renal (44.9%), hematologic (31.0%), respiratory (26.6%), neurologic (26.3%) and hepatic (21.6%).

The prevalence of IAH and ACS was 82.1% and 23.2%, respectively. Overall, mean value of maximum IAP measurements was 16.0±5.7 mmHg. The IAH grade distribution of maximum and mean IAP during ICU stay are, respectively, shown in Figs 2 and 3, and compared in S1 Fig. Measurements of IAP were recorded on a thrice daily basis in 56 patients with a median of 18 [8, 30] measurements per patient and a total of 1,189 measurements. For the remaining 39 patients, single mean and maximum values were reported without reference to the number of measurements performed.

Risk factors associated with IAH in univariable analysis were higher serum sodium concentration (p = 0.02), higher Glasgow Coma Scale (GCS) (p = 0.03), and higher West-Haven (WH) hepatic encephalopathy score (p = 0.04), while alcoholic etiology of cirrhosis (p = 0.06)

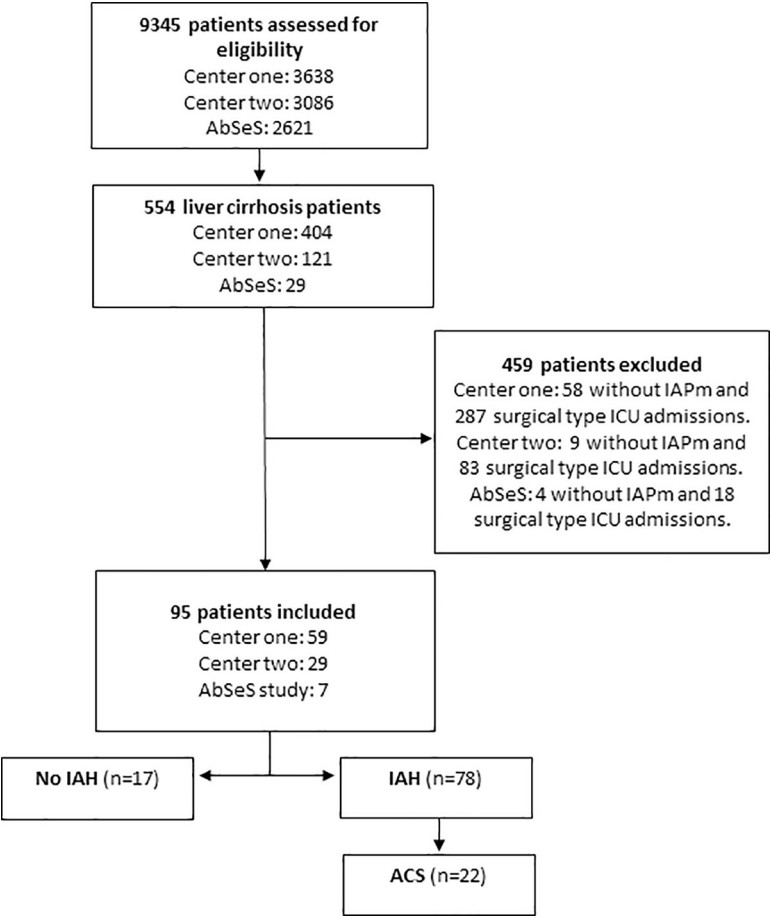

**Fig 1. Patient flowchart.** Patients with liver cirrhosis were considered eligible and were excluded if surgical type ICU admission or absence of IAPm. Abbreviations: AbSeS–Abdominal Sepsis Study: Epidemiology of Etiology and Outcome; ACS–Abdominal Compartment Syndrome; ICU–Intensive Care Unit; IAH–Intra-abdominal Hypertension; IAPm–Intra-Abdominal Pressure measure.

and PaO2/FiO2 (p = 0.1) nearly reached significant association (Table 2). Multivariable analysis (excluding GCS due to preference for WH encephalopathy score in the cirrhotic patient) revealed that alcoholic cirrhosis (p = 0.01), higher WH scores (p = 0.01), and higher PaO2/FiO2 ratio (p = 0.02) were independently associated with the presence of IAH (S1 Table).

Abdominal compartment syndrome was significantly associated with higher MELD and MELD-Na scores (respectively, p = 0.03 and p = 0.04), and precipitant event (p = 0.03) compared to other patients with simple IAH. White blood cell count (p = 0.08) nearly reached significant association with ACS (Table 2). Multivariate analysis (n = 71), including the aforementioned variables, showed that the precipitant event "infection" (p = 0.048) was the only independent risk factor for ACS.

In-ICU mortality rate was 46.3% (n = 44), and overall, 28-day mortality was 52.6% (n = 50). Non-surviving patients had significantly higher maximum and mean IAP (p = 0.02 and p = 0.04, respectively) (Table 1) and patients with ACS had significantly higher 28-day mortality when compared to those with lower IAH grades I–II (77% versus 46%, respectively, p = 0.02) (Table 2). IAH grade distribution and respective mortality rates for either maximum or mean IAP values are separately shown in Figs 2 and 3.

**Table 1. Baseline characteristics of liver cirrhosis patients in intensive care and 28-day mortality.**

| | All patients (n = 95) | | | Non-survivors (n = 44) | | | Survivors (n = 50) | | | p |
|---|---|---|---|---|---|---|---|---|---|---|
| Age (years) | 56.7 | ± | 1.3 | 57.2 | ± | 11.0 | 56.3 | ± | 10.0 | 0.5 |
| Male gender (n, %) | 75 | | (79) | 41 | | (93) | 34 | | (68) | 0.9 |
| Etiology of liver cirrhosis (n, %) [a] | | | | | | | | | | 0.4 |
| Alcohol | 43 | | (45) | 26 | | (59) | 17 | | (34) | |
| Alcohol plus HCV | 6 | | (6) | 3 | | (7) | 6 | | (12) | |
| HCV | 6 | | (6) | 3 | | (7) | 3 | | (6) | |
| Other | 37 | | (39) | 18 | | (41) | 19 | | (38) | |
| Precipitating event (n, %) [b] | | | | | | | | | | 0.6 |
| Infection | 25 | | (26) | 14 | | (31) | 11 | | (22) | |
| Bleeding | 20 | | (21) | 12 | | (27) | 8 | | (16) | |
| Other | 50 | | (53) | 24 | | (55) | 26 | | (52) | |
| CCS (n = 64) | 5.4 | ± | 2.1 | 5.6 | ± | 2.1 | 5.1 | ± | 2.1 | 0.4 |
| MELD (n = 87) | 26.2 | ± | 9.9 | 30.4 | ± | 30.4 | 21.6 | ± | 7.5 | <0,001 |
| MELD-Na (n = 87) | 27.7 | ± | 9 | 31.4 | ± | 9.0 | 23.7 | ± | 7.1 | <0,001 |
| APACHE II (n = 88) | 25.3 | ± | 10.1 | 28.2 | ± | 11.3 | 22.1 | ± | 7.7 | 0.004 |
| SAPS II (n = 89) | 48.5 | ± | 15.3 | 54.7 | ± | 15.1 | 41.5 | ± | 12.5 | <0,001 |
| CLIF-SOFA (n = 87) | 12.8 | ± | 3.6 | 13.7 | ± | 3.7 | 11.8 | ± | 3.3 | 0.01 |
| SOFA (n = 83) | 11.3 | ± | 3.4 | 11.7 | ± | 3.4 | 10.8 | ± | 3.4 | 0.2 |
| Organ failure (n = 87) | 2.2 | ± | 1.2 | 2.5 | ± | 1.3 | 1.9 | ± | 1.0 | 0.01 |
| Ascites (n = 64) (n, %) | 59 | | (92) | 32 | | (89) | 27 | | (96) | 0.4 |
| West-Haven score (Q1-Q3) | 1 | | (0, 3) | 1.0 | | (0, 3) | 0.0 | | (0, 2) | 0.2 |
| GCS (Q1-Q3) (n = 88) | 14 | | (8, 15) | 14.0 | | (8, 15) | 15.0 | | (8, 15) | 0.2 |
| Ammonia (mmol/L)(Q1-Q3)(n = 34) | 153 | | (104, 237) | 155 | | (120, 221) | 144 | | (74, 258) | 0.3 |
| Hematocrit (%) (n = 59) | 24.4 | ± | 5.9 | 23.8 | ± | 5.8 | 25.1 | ± | 6.1 | 0.4 |
| Leucocytes (10 x $10^9$/mL) (n = 94) | 13.7 | ± | 8,6 | 15.9 | ± | 9.5 | 11.3 | ± | 7.0 | 0.01 |
| Platelets (10 x $10^9$/mL) (n = 85) | 118 | ± | 91 | 129 | ± | 80.1 | 106 | ± | 101 | 0.2 |
| INR (n = 87) | 2.4 | ± | 1.3 | 2.9 | ± | 1.6 | 1.9 | ± | 0.6 | <0,001 |
| Bilirubin (mg/dl) (n = 88) | 8.5 | ± | 9.6 | 11.3 | ± | 10.6 | 5.4 | ± | 7.3 | 0.003 |
| Creatinine (mg/dl) (n = 88) | 2.1 | ± | 1.5 | 2.3 | ± | 1.5 | 1.9 | ± | 1.5 | 0.2 |
| Sodium (mEq/L) (n = 95) | 136 | ± | 7.9 | 136 | ± | 7.8 | 137 | ± | 8 | 0.5 |
| C-reactive protein (mg/L) (n = 58) | 60.7 | ± | 60.1 | 55.8 | ± | 63.4 | 63.3 | ± | 58.9 | 0.6 |
| Lactate (mmol/l) (n = 58) | 4.1 | ± | 4.3 | 6.6 | ± | 6.1 | 3.5 | ± | 3.4 | 0.004 |
| PaO2/FiO2 (n = 83) | 285 | ± | 111 | 311 | ± | 130.3 | 262 | ± | 99 | 0.06 |
| Vital organ support (n, %) [c] | 81 | | (85) | 42 | | (84) | 39 | | (87) | 0.8 |
| Vasopressors | 68 | | (72) | 37 | | (74) | 31 | | (69) | 0.7 |
| IMV | 63 | | (66) | 35 | | (70) | 28 | | (62) | 0.5 |
| RRT | 19 | | (20) | 12 | | (24) | 7 | | (16) | 0.4 |
| Maximum IAP (mmHg) | 16 | ± | 5.7 | 17.3 | ± | 6.3 | 14.6 | ± | 4.7 | 0.02 |
| Mean IAP (mmHg) | 12.1 | ± | 4.1 | 13.0 | ± | 4.8 | 11.2 | ± | 2.8 | 0.03 |
| ICU LOS | 10.8 | ± | 11.3 | 8.7 | ± | 8.7 | 13.1 | ± | 13.4 | 0.07 |

Number of observations (n) equals 95 and are presented in mean and SD unless otherwise stated.

[a] p value is provided for the comparison of "alcohol alone plus combined" versus all other liver cirrhosis etiologies.

[b] p value is provided for the comparison of bleeding or infection versus all other precipitating events.

[c] Vital organ support refers to single or combined vasopressor, IMV or RRT during the entire ICU stay.

Abbreviations: ACS—abdominal compartment syndrome; APACHE II—Acute Physiology and Chronic Health Evaluation II; CCS—Charlson Comorbidity Score; CLIF—Chronic Liver Failure; GCS—Glasgow Coma score; HCV—hepatitis C virus; IAP—intra-abdominal pressure; IAH—intra-abdominal hypertension; IAP—intra-abdominal pressure; ICU—intensive care unit; INR—international normalization ratio; Q1- 1st quartile; Q3 - 3rd quartile; LOS—length-of-stay; MELD—Model for End Stage Liver Disease; MELD-Na—Model for End Stage Liver Disease Sodium; SAPS II—Simplified Acute Physiology Score II; SD—standard deviation; SOFA—Sequential Organ Failure Assessment.

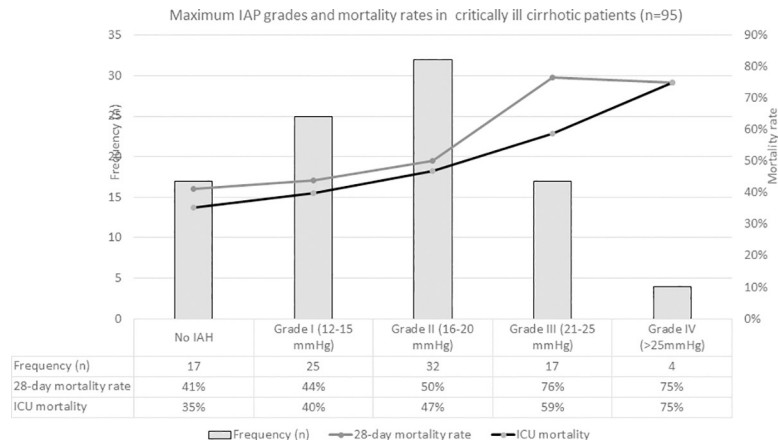

**Fig 2. Distribution of IAH grades for maximum intra-abdominal pressure and mortality rates in critically ill liver cirrhotic patients.** Abbreviations: IAH–Intra-abdominal hypertension; ICU–intensive care unit.

Univariate analysis of 28-day mortality revealed that non-survivors had significantly worse clinical severity scores, except for the SOFA and Charlson Comorbidity Index, and significantly elevated values for laboratory variables bilirubin, INR, lactate, and WBC at admission, with PaO2/FiO2 nearly reaching significant association.

Maximum IAP was associated with 28-day mortality when adjusted for SAPS II score (OR = 1.11; 95% CI: 1.01–1.22; p = 0.02 for n = 89) but not for MELD nor MELD-Na (respectively, OR = 1.06; 95% CI: 0.97–1.15; p = 0.2; and OR = 1.07; 95% CI: 0.98–1–16; p = 0.1 for n = 87).

Multivariable analysis including lactate, WBC, PaO2/FiO2, maximum IAP, and MELD score (excluding bilirubin and INR as these variables are included in MELD score) identified MELD (p = 0.001), WBC (p = 0.03), PaO2/FiO2 (p = 0.03), and lactate (p = 0.04) as independent predictors of 28-day mortality (S2 Table).

Classification of OF at ICU admission, with matching mortality rates, is depicted in S2 Fig.

There was no significant correlation found between mean IAP or maximum IAP and ICU length-of-stay.

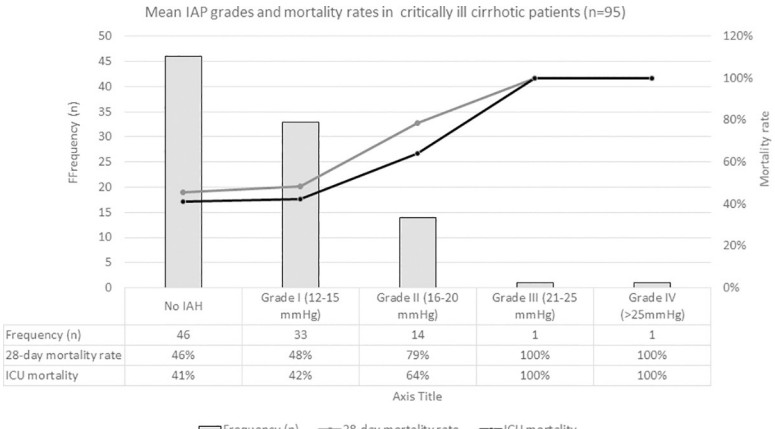

**Fig 3. Distribution of IAH grades for mean intra-abdominal pressure and mortality rates in critically ill liver cirrhotic patients.** Abbreviations: IAH–Intra-abdominal hypertension; ICU–intensive care unit.

Liver transplantation, although not a reason for ICU admission, was eventually performed in 8 (13.6%) of 59 patients during a 28-day follow-up.

## Discussion

This is the first multicentric study to analyze the prevalence and outcomes of IAH and ACS in cirrhotic patients admitted in the ICU with acute medical illness.

The prevalence of IAH and ACS was very high, affecting more than three quarters and nearly one quarter, respectively, of this critically ill population with decompensated liver disease, admitted mainly due to sepsis or upper gastrointestinal bleeding.

Our results show higher prevalence of IAH/ACS in cirrhotic patients when compared to previous studies in mixed populations of critical care patients, reporting a prevalence of IAH between 32.1% to 58.8% and of ACS from 4.2% to 8.2% [3–5]. In a multicentric IAP study of a mixed population of 265 patients, IAH was present in 32.1% and ACS in 4.2% of patients on ICU admission day [4]. The IROI Study, a multicentric prospective study of 491 mixed ICU patients, demonstrated IAH prevalence in 48.9% and ACS in 6.3% of patients during the first 2 weeks of the ICU stay [5]. However, the prevalence of IAP specifically in the critically ill cirrhotic patient has been earlier addressed in a post hoc analysis from a single center randomized controlled trial for the use of low-dose hydrocortisone in septic shock [22, 23]. In their study, Al-Dorzi et al. observed 61 cirrhotic patients over the first week in the ICU and described IAH in 97% and ACS in 39% of them. Ascites was present in nearly all patients (95.1%) and, as recognized by the authors, they very likely overestimated the IAH/ACS prevalence since intra-vesical pressure measurements were performed with vesical instillation of a large volume (50–100 mL), and the zero-pressure reference point was set at symphysis pubis. Current guidelines recommend an injection of 25 mL of sterile saline to prevent bladder over-distension and zero-pressure reference set at the phlebostatic axis in the mid-axillary line [1]. Even so, and considering a possible patient selection bias, the prevalence of IAH was remarkably high, in line with our findings.

The present study suggests interrelations between etiology and severity of hepatic disease, degree of organ failures and IAH, as we observed IAH being independently associated with alcoholic cirrhosis, hepatic encephalopathy WH score, and respiratory dysfunction (PaO2/FiO2 ratio). This is in accordance with previous studies where liver dysfunction (likely due to the formation of ascites), severe acute illness (APACHE II ≥18 points), respiratory dysfunction (mechanical ventilation with positive end-expiratory pressure ≥7 cm), abdominal distension, ileus, positive fluid balance, obesity (body mass index >27 kg/m$^2$), and abdominal surgery increase the likelihood of IAH development [4, 5]. Furthermore, our findings showed that infection as a precipitant event of decompensated liver disease was an independent risk factor for the presence of ACS among those with IAH. Though it could be speculated that this finding may have reflected spontaneous bacterial peritonitis or volume resuscitation in the septic shock patient with increased ascites formation or, perhaps, pneumonia with IMV and high PEEP level transmitted to the abdomen, this could not be clarified.

Finding that IAH was associated with higher hepatic encephalopathy scores deserves attention. Few studies have reported a correlation between IAH and increased intracranial pressure (ICP), even at low levels of IAP.[21] Furthermore, reversible disruption of the blood-brain barrier and increased ischemic mediators, indicating cerebral ischemia due to IAH, have been demonstrated [24–29]. Two distinct pathways by which IAP could be transmitted to the central nervous system have been proposed, according to the Monro-Kellie hypothesis: first, through the venous plexus of the spinal canal and the intracranial veins backflow, and second, due to cranial excursion of the diaphragm causing elevated intrathoracic pressure as well as

**Table 2. Intra-abdominal hypertension in the critically ill cirrhotic patients.**

| | No IAH (n = 17) | | | IAH (n = 78) | | | | IAH (n = 56) | | | ACS (n = 22) | | | |
|---|---|---|---|---|---|---|---|---|---|---|---|---|---|---|
| | IAP < 12 mmHg | | | IAP ≥ 12 mmHg | | | p | IAP [12–20 [mmHg | | | IAP ≥ 20 mmHg | | | p |
| Age (years) | 56.5 | ± | 10.1 | 56.8 | ± | 10.6 | 0.90 | 56.7 | ± | 10.8 | 57.2 | ± | 10.5 | 0.86 |
| Male gender (n, %) | 15 | | (88) | 60 | | (77) | 0.51 | 42 | | (75) | 18 | | (82) | 0.77 |
| Etiology of cirrhosis (n, %) [a] | | | | | | | 0.2 | | | | | | | 0.6 |
| Alcohol | 10 | | (59) | 33 | | (42) | | 24 | | (43) | 9 | | (41) | |
| Alcohol plus HCV | 3 | | (18) | 6 | | (7,7) | | 4 | | (7,1) | 2 | | (9,1) | |
| HCV | 0 | | (0) | 6 | | (7,7) | | 3 | | (5,4) | 3 | | (14) | |
| Other | 4 | | (24) | 33 | | (42) | | 25 | | (45) | 8 | | (36) | |
| Precipitating event (n, %) [b] | | | | | | | 0.85 | | | | | | | 0.03 |
| Infection | 4 | | (24) | 21 | | (27) | | 12 | | (21) | 9 | | (41) | |
| Bleeding | 3 | | (18) | 17 | | (22) | | 10 | | (18) | 7 | | (32) | |
| Other | 10 | | (59) | 40 | | (51) | | 34 | | (61) | 6 | | (27) | |
| CCS (n = 64) | 4.9 | ± | 2.0 | 5.5 | ± | 2.1 | 0.40 | 5.3 | ± | 2.3 | 5.8 | ± | 2.0 | 0.45 |
| MELD (n = 87) | 26.7 | ± | 9.0 | 26.0 | ± | 10.2 | 0.81 | 24.4 | ± | 9.7 | 30.2 | ± | 10.4 | 0.03 |
| MELD-Na (n = 87) | 28.9 | ± | 8.1 | 27.4 | ± | 9.2 | 0.55 | 26.0 | ± | 8.8 | 31.0 | ± | 9.5 | 0.04 |
| APACHE II (n = 88) | 27.5 | ± | 15.2 | 24.8 | ± | 8.7 | 0.33 | 24.7 | ± | 9.1 | 24.9 | ± | 7.7 | 0.95 |
| SAPS II (n = 89) | 48.2 | ± | 12.3 | 48.5 | ± | 16.0 | 0.94 | 48.1 | ± | 17.2 | 49.5 | ± | 12.7 | 0.75 |
| CLIF-SOFA (n = 87) | 12.8 | ± | 3.1 | 12.8 | ± | 3.8 | 0.98 | 12.9 | ± | 3.9 | 12.6 | ± | 3.5 | 0.74 |
| SOFA (n = 83) | 10.4 | ± | 2.8 | 11.4 | ± | 3.5 | 0.28 | 11.7 | ± | 3.5 | 10.8 | ± | 3.4 | 0.34 |
| Organ failure (n = 87) | 2.2 | ± | 1.4 | 2.2 | ± | 1.2 | 0.94 | 2.3 | ± | 1.1 | 2.1 | ± | 1.3 | 0.62 |
| Ascites (n = 64) (n, %) | 11 | | (92) | 48 | | (92) | 1.00 | 30 | | (94) | 18 | | (90) | 0.63 |
| West-Haven score (Q1-Q3) | 0.0 | | (0, 1) | 1.0 | | (0, 3) | 0.04 | 1.0 | | (0, 3) | 1.0 | | (0, 3) | 0.34 |
| GCS (Q1-Q3) (n = 88) | 15 | | (14, 15) | 14 | | (7, 15) | 0.03 | 14 | | (7, 15) | 15 | | (9, 15) | 0.14 |
| Ammonia (mmol/L)(Q1-Q3)(n = 34) | 150 | | (94, 235) | 155 | | (106, 243) | 0.97 | 157 | | (92, 243) | 147 | | (117, 213) | 0.78 |
| Hematocrit (%) (n = 59) | 24.0 | ± | 6.7 | 24.4 | ± | 5.8 | 0.84 | 23.5 | ± | 4.7 | 26.1 | ± | 7.1 | 0.13 |
| Leucocytes (10 x $10^9$/mL) (n = 94) | 14.4 | ± | 9.7 | 13.5 | ± | 8.4 | 0.69 | 12.4 | ± | 8.1 | 16.1 | ± | 8.9 | 0.08 |
| Platelets (10 x $10^9$/mL) (n = 85) | 131 | ± | 136 | 116 | ± | 81 | 0.56 | 111 | ± | 78 | 127 | ± | 89 | 0.48 |
| INR (n = 87) | 2.5 | ± | 1.2 | 2.4 | ± | 1.4 | 0.73 | 2.2 | ± | 1.1 | 2.7 | ± | 1.9 | 0.17 |
| Bilirubin (mg/dl) (n = 88) | 7.4 | ± | 7.4 | 8.7 | ± | 10.1 | 0.63 | 7.7 | ± | 9.4 | 11.4 | ± | 11.4 | 0.16 |
| Creatinine (mg/dl) (n = 88) | 2.0 | ± | 1.4 | 2.1 | ± | 1.6 | 0.78 | 2.0 | ± | 1.6 | 2.5 | ± | 1.6 | 0.28 |
| Sodium (mEq/L) (n = 95) | 132 | ± | 9 | 137 | ± | 7 | 0.02 | 137 | ± | 8 | 136 | ± | 6 | 0.53 |
| C-reactive protein (mg/L) (n = 58) | 73 | ± | 65 | 56 | ± | 60 | 0.36 | 51 | ± | 49 | 70 | ± | 83 | 0.23 |
| Lactate (mmol/l) (n = 58) | 5.1 | ± | 5.2 | 5.1 | ± | 5.2 | 0.96 | 5.1 | ± | 5.2 | 5.1 | ± | 5.5 | 0.97 |
| PaO2/FiO2 (mmHg) (n = 83) | 242 | ± | 112 | 298 | ± | 119 | 0.10 | 291 | ± | 126 | 315 | ± | 100 | 0.45 |
| Vital organ support (n, %) [c] | 15 | | (88) | 66 | | (85) | 1.00 | 51 | | (91) | 15 | | (68) | 0.31 |
| Vasopressors | 11 | | (65) | 57 | | (73) | 0.56 | 43 | | (77) | 14 | | (64) | 0.27 |
| IMV | 12 | | (71) | 51 | | (65) | 0.78 | 40 | | (71) | 11 | | (50) | 0.11 |
| RRT | 3 | | (18) | 16 | | (21) | 1.00 | 14 | | (25) | 2 | | (9,1) | 0.21 |
| Maximum IAP (mmHg) | 7.9 | ± | 2.0 | 17.8 | ± | 4.6 | <0,001 | 15.4 | ± | 2.5 | 23.8 | ± | 3.2 | <0,001 |
| Mean IAP (mmHg) | 7.2 | ± | 2.0 | 13.2 | ± | 3.5 | <0,001 | 12.3 | ± | 2.6 | 15.7 | ± | 4.4 | <0,001 |
| 28-day mortality | 7 | | (41) | 43 | | (55) | 0.42 | 26 | | (46) | 17 | | (77) | 0.02 |
| ICU mortality | 6 | | (35) | 38 | | (49) | 0.42 | 24 | | (43) | 14 | | (64) | 0.13 |
| ICU LOS (days) | 9.4 | ± | 10.8 | 11.1 | ± | 11.5 | 0.60 | 11.4 | ± | 13.0 | 10.3 | ± | 6.1 | 0.71 |

Number of observations (n) equals 95 and values are presented as mean (± SD) unless otherwise stated.

[a] p value is provided for the comparison of "alcohol alone plus combined" versus all other liver cirrhosis etiologies.

[b] p value is provided for the comparison of bleeding or infection versus all other precipitating events.

[c] Vital organ support refers to single or combined vasopressor, IMV or RRT during the entire ICU stay.

Abbreviations: ACS—abdominal compartment syndrome; APACHE II—Acute Physiology and Chronic Health Evaluation II; CCS—Charlson Comorbidity Score; CLIF —Chronic Liver Failure; GCS—Glasgow Coma score; HCV—hepatitis C virus; IAP—intra-abdominal pressure; IAH—intra-abdominal hypertension; IAP—intra-abdominal pressure; ICU—intensive care unit; INR—international normalization ratio; Q1- 1st quartile; Q3 – 3rd quartile; LOS—length-of-stay; MELD—Model for End Stage Liver Disease; MELD-Na—Model for End Stage Liver Disease Sodium; SAPS II—Simplified Acute Physiology Score II; SD—standard deviation; SOFA— Sequential Organ Failure Assessment.

augmented central venous pressure decreasing venous drainage from the central nervous system via the jugular system [24, 30, 31]. All these mechanisms could represent an additional pathophysiologic rationale for hepatic encephalopathy in liver disease patients with increased IAP [24–31].

A higher PaO2/FiO2 was, strikingly, associated with IAH. The reason for this observation was not obvious to us and it could be speculated that higher levels of PEEP may have been used in these mechanically ventilated patients with IAH, thus increasing PaO2, although, available data could not confirm this.

Alcoholic cirrhosis was also associated with IAH, but the fact that over half of the study population had alcohol related cirrhosis and the remaining etiologies were either scarce in number or unspecified prevented us from elaborating on this aspect.

A high mortality rate was observed and comparable with other studies in critically ill liver cirrhosis patients showing short-term mortality rates ranging from 34.9% to 71.9% [32–34].

Increased IAP was associated with higher 28-day mortality, particularly with ACS. The severity of IAH was progressively associated with mortality, which was similarly high at IAH Grades III and IV. This observation strengthens the argument for merging these grades together, as suggested by Blaser et al. [5].

Mean IAP presented higher mortality rates than maximum IAP for matching grades of IAH. It could be speculated that the deleterious clinical impact of sustained (mean) IAP, more so than episodical (maximum) IAP, plays a predominant role in the ultimate effect and highlights the importance of the duration of IAH and time-dependence of its pathological effects on organ dysfunction [2]. Our observations also revealed that the distribution of mean IAP was mostly restricted to grade I-II, in contrast with maximum IAP values (S1 Fig). This difference could have been partly due to the clinical management of IAH reflecting a sustained reduction of IAP throughout the ICU stay. Furthermore, the broader grade distribution of maximum IAP values could indicate a better marker of severity of disease.

Multivariable analysis of 28-day mortality identified MELD score, WBC count, Pa/FiO2 ratio, and lactate concentration as independent risk factors, confirming other studies [17, 33, 35, 36].

Although the IROI study reported the presence and severity of IAH as an independent risk factor for increased mortality in a mixed population of critical patients, our study did not observe this [5]. Similarly, IAH was reported to be associated with renal failure and RRT in septic shock liver disease patients and in those requiring longer ventilatory and vasopressor support [17]. However, we did not observe such associations. A likely explanation is the selected population since the prevalence of IAH in our study was notably high. Further research is needed to confirm these findings.

Clinical practice guidelines for the critically ill patient with IAH/ACS recommend a stepwise medical management algorithm to reduce and maintain IAP ≤15 mmHg. Importantly, ascites stands out as a fundamental common feature in critical liver disease patients with IAH. Studies in intensive care have reported the safety of large volume paracentesis, thus potentially lowering IAP, with beneficial effects on hepatosplenic blood flow as well as respiratory and renal functions [37–41]. Particular relevance should be assigned to the clinical management of cirrhotic patient with ascites, given that paracentesis can treat and potentially prevent the recurrence of IAH and ACS during an ICU stay.

Some noteworthy limitations in this study precluded more accurate reporting and interpretation of results. These include valuable variables that were not available, such as abdominal perfusion pressure, fluid balance, IAP trends during ICU stay. Data regarding the impact of specific therapeutic interventions aimed at lowering IAP, importantly, abdominal paracentesis, was not available and fell out of the scope of this study. Possible selection and sampling biases

were minimized due to the multicentric collection of data at each study site and the reasonably sized patient cohort. External validation of our results is required, although we believe them to be generalizable given the multicentric design of the study and the reasonable number of patients.

The randomized controlled trial on this subject (COPPTRIAHL, NCT04322201) is underway with the intent to control IAH for improved clinical outcomes and may provide future insights on this topic [42].

## Conclusions

Our study demonstrates a very high prevalence of IAH/ACS in the critically ill liver cirrhotic patient with acute medical illness when compared to other populations of mixed intensive care patients. Increased IAP was associated with severity of disease and adverse outcomes and independent risk factors for IAH were alcoholic cirrhosis, hepatic encephalopathy and PO2/FiO2 ratio, as well as infection for ACS. Particular relevance should be given to early diagnosis, treatment, and prevention of IAH and ACS as it might improve outcomes in the liver cirrhosis patient in the ICU.

## Supporting information

**S1 Fig. Comparison of IAH grades for maximum and mean IAP in critically ill liver cirrhotic patients.** Abbreviations: IAH–Intraabdominal Hypertension; IAP–Intra-abdominal Pressure.
(TIF)

**S2 Fig. Number of organ failures at ICU admission and mortality rates in liver cirrhotic patients.** Abbreviations: OF–organ failure; ICU–intensive care unit.
(TIF)

**S1 File. Collaborators AbSeS study.** List of collaborators from the AbSeS study.
(DOCX)

**S1 Table. Multivariable logistic regression for association with intra-abdominal hypertension.** Multivariable backward stepwise logistic regression including alcoholic cirrhosis, PaO2/FiO2, West-Haven hepatic encephalopathy score and Sodium in the initial step.
(DOCX)

**S2 Table. Multivariable logistic regression for 28-day mortality risk factor analysis in critically ill liver cirrhotic patients.** Multivariable backward stepwise logistic regression (n = 83), lactate, WBC, PaO2/FiO2, maximum IAP and MELD score in the initial step (bilirubin and INR are included in MELD score and were therefore excluded from this analysis). Abbreviations: FiO2—fraction of inspired oxygen; IAP- intra-abdominal pressure; INR—international normalization ratio; MELD–Model for End-stage Liver Disease; PaO2—partial arterial oxygen pressure; WBC—white blood cell.
(DOCX)

**S1 Data.**
(SAV)

## Acknowledgments

The authors of this study would like to acknowledge Dr. Nuno Germano, the Intensive Care Medicine nursing and medical staff at Hospital de Curry Cabral and Tartu University

Hospital, the ESICM Trials Group Project collaborators in the "Abdominal Sepsis Study: Epidemiology of Etiology and Outcome" (S1 Fig) and Centro de Investigação at Centro Hospitalar Universitário Lisboa Central for their contribution.

## Author Contributions

**Conceptualization:** Rui Pereira, Rui Perdigoto, Paulo Marcelino, Faouzi Saliba, Stijn Blot, Joel Starkopf.

**Data curation:** Rui Pereira.

**Formal analysis:** Rui Pereira.

**Investigation:** Rui Pereira, Maria Buglevski, Stijn Blot, Joel Starkopf.

**Methodology:** Rui Pereira, Joel Starkopf.

**Project administration:** Rui Pereira.

**Supervision:** Rui Pereira.

**Validation:** Rui Perdigoto, Joel Starkopf.

**Writing – original draft:** Rui Pereira.

**Writing – review & editing:** Rui Pereira, Maria Buglevski, Rui Perdigoto, Paulo Marcelino, Faouzi Saliba, Stijn Blot, Joel Starkopf.

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
