## [Decision Letter · Decision Letter 0]

17 Mar 2021

PONE-D-21-05197

Intra-Abdominal Hypertension and Abdominal Compartment Syndrome in the critically ill liver cirrhotic patient - prevalence and clinical outcomes. A multicentric retrospective cohort study in intensive care.

PLOS ONE

Dear Dr. Pereira,

Thank you for submitting your manuscript to PLOS ONE. After careful consideration, we feel that it has merit but does not fully meet PLOS ONE’s publication criteria as it currently stands. Therefore, we invite you to submit a revised version of the manuscript that addresses the points raised during the review process.

We look forward to receiving your revised manuscript.

Kind regards,

Aleksandar R. Zivkovic

Academic Editor

PLOS ONE

3. Please include your tables as part of your main manuscript and remove the individual files. Please note that supplementary tables should remain as separate "supporting information" files.

Reviewers' comments:

Reviewer's Responses to Questions

**Comments to the Author**

1. Is the manuscript technically sound, and do the data support the conclusions?

Reviewer #1: Partly

Reviewer #2: Yes

2. Has the statistical analysis been performed appropriately and rigorously? 

Reviewer #1: Yes

Reviewer #2: I Don't Know

3. Have the authors made all data underlying the findings in their manuscript fully available?

Reviewer #1: Yes

Reviewer #2: Yes

4. Is the manuscript presented in an intelligible fashion and written in standard English?

Reviewer #1: Yes

Reviewer #2: No

5. Review Comments to the Author

Reviewer #1: I would like to thank the authors for their submission, on an important topic in critical care medicine. As correctly commented upon in the article, although there are a number of well recognised risk factors for the development of IAH and ACS in critically ill patients, the data on many of these risk factors is extremely sparse. As such, the work examining one of these risk factors in greater depth (Hepatic impairment/cirrhosis) is welcomed. I would like to see this work published, but there are a number of comments and recommendations which I would make in order to make this submission more accessible for the readership. I have attached specific comments.

Introduction:

• “The reported prevalence of IAH/ACS in mixed medical and surgical ICU patients reach 59% and 8%, respectively.” – It is unclear as to what these figures are referring. The prevalence of IAH and ACS are two different entities. It would be helpful here to define IAH and ACS. The prevalence of IAH in this population is much higher than that of ACS. As per the WSACS consensus statement, IAH is an intra-abdominal pressure >12mmHg, whilst ACS is >20mmHg with organ dysfunction. This should be clearly define in the text.

• Although there is discussion of the treatment of ACS by paracentesis, it would be useful to clarify what the clinical sequelae are of the condition, For example, renal impartment, ileus and intestinal failure may all result from IAH and should be included in order to place the article in the clinical context.

• What is the operative intervention rate and mortality rate of IAH/ACS?

Materials and Methods

• It is useful to include multicentre work in such studies and this is to be recognised as a valuable approach. However there appears to be a discrepancy in the length of time over which the data was collected (October 2016-2019 in the first centre, 2010-2018 in the second). Why is this? Was this due to a change in practice at these centres or due to data availability?

• “The Institutional Ethics Committees waived the need for individual informed consent for this observational study.” – I am assuming the ethics committees for the two institutions named, could you clarify this in the text?

• You excluded patients if they had an ICU admission for ‘surgical reasons’. Is this purely abdominal surgery? This would have a different confounding effect on the prevalence of IAH compared to, for example, orthopaedic surgery or trauma. This needs clarification.

• The overall stuctures of the methods section is a little diiffiicult to follow. I would consider restructuring it into baseline variables, IAP assessment, clinical management, statistical analysis.

• The methods of statistical analysis appear appropriate.

Results

• I am not clear on the data in figure one. You have stated 554 eligible patients, but then in the next step on the flowchart excluded 459 patients. Would it not be better to say than 554 developed IAH? Or do you mean there were 554 cirrhotic patients? This really isn’t clear and needs to be amended.

• Have you considered an analysis of the timing of IAH measurement? For instance, it would be helpful to understand if mean IAP was related to the length of stay in the ICU.

• It is interesting that alcoholic cirrhosis was an independent risk factor. Did you undertake an analysis also of the underlying aetiology of ICU admission for worsening of cirrhosis? I would also like to know how many patients had paracentesis, as this would likely be a significant factor in prevention of ACS.

Discussion

• I think the conclusions you have drawn from your results are reasonable. However, it should be noted that the lack of abdominal perfusion pressure is a major limitation. Presumably this could have bene obtained, and it may be worth looking at these patients again if the mean arterial pressure is available (although I understand that not all patients will have invasive blood pressure monitoring.

• Certainly, your results are in line with other studies looking at cirrhosis as a risk factor for abdominal compartment syndrome, which as you have quite rightly identified has a significantly higher prevalence in this population.

Overall this is a useful study looking at an important subset of patients developing IAH and ACS in critical care, and certainly it would contribute to the knowledge of the condition in this setting. I would therefore like to see it published. However, I believe there are some major amendments needed to ensure this paper is suitable for publication.

Reviewer #2: - What is the standard IAP measurement in this study? Is it the same technique in both centers in this study?

- Can you define the accurate study month in the second center?

- What is the frequency of IAP measurement? and How to calculate the data (cmH2O to mmHg) in this study?

- IQR should be represent in (Q1, Q3) not the range.

- This study reported very high prevalence of IAH. It may from the measurement method or recording technique. Please define the accurate method.

- In tables, the classification of the data should be made to make it easier to read such as demographic data, cirrhosis parameters and scores, clinical signs, laboratory, measurement data, and result.

- In patient flowchart, this study defines as a cohort study, the analyzed cases should be classified in to groups which the investigator would like to compare (IAH, ACS, no IAH?).

- In figures, categorized of IAH/ACS is based on the maximum IAP, not the mean IAP that why the frequency of the data of figure 2 and 3 is not same. Please revise these figure.

- You can make this article easier to read by reducing paragraphs and unnecessary words.

6. PLOS authors have the option to publish the peer review history of their article (what does this mean?). If published, this will include your full peer review and any attached files.

Reviewer #1: **Yes: **Mr Nathan Tyson

Reviewer #2: No

---

## [Author Response · Author response to Decision Letter 0]

26 Apr 2021

Thank you for your feedback and the opportunity to improve this manuscript.

The "Responses to Reviewers" file answers all the questions posed by the editor and reviewers.

I hope it proves to be adequate. Should there be any further issues I will be glad to comply.

Looking forward to hear from you.

Kind regards.

Rui Pereira

---

## [Editor Report · Decision Letter 1]

28 Apr 2021

Intra-Abdominal Hypertension and Abdominal Compartment Syndrome in the critically ill liver cirrhotic patient - prevalence and clinical outcomes. A multicentric retrospective cohort study in intensive care.

PONE-D-21-05197R1

Dear Dr. Pereira,

We’re pleased to inform you that your manuscript has been judged scientifically suitable for publication and will be formally accepted for publication once it meets all outstanding technical requirements.

Kind regards,

Aleksandar R. Zivkovic

Academic Editor

PLOS ONE

---

## [Editor Report · Acceptance letter]

3 May 2021

PONE-D-21-05197R1 

Intra-Abdominal Hypertension and Abdominal Compartment Syndrome in the critically ill liver cirrhotic patient - prevalence and clinical outcomes. A multicentric retrospective cohort study in intensive care. 

Dear Dr. Pereira:

I'm pleased to inform you that your manuscript has been deemed suitable for publication in PLOS ONE. Congratulations! Your manuscript is now with our production department. 

Kind regards, 

on behalf of

Dr. Aleksandar R. Zivkovic 

Academic Editor

PLOS ONE